# Numerical Analysis Results of Debonding Damage Effects for an SHM System Application on a Typical Composite Beam

**Gianluca Diodati** [1], **Assunta Sorrentino** [1,*], **Lorenzo Pellone** [2], **Antonio Concilio** [3], **Monica Ciminello** [3], **Gianvito Apuleo** [4], **Shay Shoham** [5], **Iddo Kressel** [5] and **David Bardenstein** [5]

1. Vibro-Acoustic Laboratory, The Italian Aerospace Research Centre (CIRA), 81043 Capua, Italy
2. Aeronautical Technologies Integration, The Italian Aerospace Research Centre (CIRA), 81043 Capua, Italy
3. Adaptive Structures Division, The Italian Aerospace Research Centre (CIRA), 81043 Capua, Italy
4. Research Division, Piaggio Aerospace Industries, 81043 Capua, Italy
5. Advanced Structural Technologies, Engineering Center, Israel Aerospace Industries (IAI), Ben Gurion International Airport, Tel Aviv 70100, Israel
* Correspondence: a.sorrentino@cira.it

**Abstract:** In the aeronautical field, the damage that occurs to a carbon-fibre-reinforced polymer (CFRP) structure analysis is a crucial point for further improving its capability and performance. In the current the state of the art, in fact, many issues are linked to the certification process more than to technological aspects. For the sake of clarity, it should be added that regulations call for technological solutions that are invasive (in terms of weight and manufacturing costs) or exploit technologies that are not fully mature. Thus, the truth is in between the above statements. One of the possible solutions to bypass this issue is the assessment of a structural health monitoring system (SHM) that is sufficiently reliable to provide a full-state representation of the structure, automatically, perhaps in real-time, with a minimum intervention of specialized technicians, and that can raise an alert for safe maintenance whenever necessary. Among the different systems that have been proposed in the scientific and technological literature, SHM systems based on strain acquisitions seem very promising: they deduce the presence of flaws by analysing the variations of the intimate response of the structure. In this context, the SHM using fibre optics, supported by a dedicated algorithm, seems to be able to translate the effects of the damage reading the strain field. This means that is necessary to have a full comprehension of the flaws' effects in terms of strain variation to better formulate a strategy aimed at highlighting these distortions. It should be remarked that each type of damage is distinct; imperfections of the bonding line are herein targeted since the quality of the latter is of paramount importance for ensuring the correct behaviour of the referred structure. This presents paper focuses on a deep investigation on the strain field peculiarities that arise after the imposition of irregularities in the adhesive region. The aim is to explore the damage dimension versus its effect on the strain map, especially when bonding connects different parts of a complex composite beam. By means of finite element method applied on a typical aeronautical beam, a parametric numerical simulation was performed in order to establish the influence of a debonding dimension on a reference strain map. This work provides evidence that these effects on strain flaw decrease the distancing itself of the damage. The knowledge of these effects can be highly helpful during the design of a preliminary phase of an SHM system in order to choose the most suitable sensor in terms of reading sensitivity error, the number to be used, and their location.

**Keywords:** damage effect; strain modulation; strain sensors; composite structures; bonding lines; FE representation; linear models; mesh density

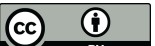

## 1. Introduction

The research on damage to composite structures is particularly vigorous in the technological and scientific world community, as this material is being increasingly used in

the construction of modern means of transport. In the aerospace sector, particularly, these investigations are motivated by specific interest. In fact, the current regulations mandate that particular attention should be devoted to the growth of damage via the insertion of a number of mechanical connections that would ensure the related flaws do not overcome the so-called critical length. For typical structures, the magnitude of this length may be estimated in 100 mm, and this necessity brings complexity, weight, and costs penalties, which significantly erode the claimed advantages of the use of composite vs. classical metal systems [1,2].

Among the different damage types that can occur in an aircraft, debonding is one of the most important since this process is the one that holds together the different components and that should substitute traditional mechanical links. Furthermore, bonding degradation over time [3] has been a subject of interest (since a structure could drastically modify its behaviour during operation) because it is one of the driving issues for a suitable maintenance plan. Several debonding criticalities may be considered: apart from the already mentioned "degradation" (or aging), there is the issue of the weak bonding, i.e., parts that partially transfer the loads to each other, [4], one of these being the so-called kissing bonds, i.e., as the interface of structural components match each other almost perfectly, this prevents any filler (the adhesive) from penetrating into the contact region. This last issue is very interesting since this phenomenon can hardly be detected with the usual non-destructive inspection (NDI) techniques such as ultrasonic wave detection systems [5], eddy current pulsed thermography [6], and the electromagnetic wave technique [7].

All of these techniques require an on-ground, off-line, periodic, and meticulous intervention of some highly specialized operator on the structure, which involves the related impacts in terms of costs and non-operational hours (grounded system). It is therefore of a relevant interest to develop solutions that could help the operator be continuously aware of the structural state and to detect poorly visible evidence that would otherwise be difficult to discern. In the literature, there are many significant examples where comparisons between NDI and SHM techniques have been discussed, such as in [8,9]. On this basis, structural health monitoring (SHM) systems are attracting a growing amount of interest despite their conceptual introduction to large-scale scientific application first occurring more than 20 years ago [10]. In fact, until now, a limited number of such architectures have been proven and can actually fly. There are many reasons for this. When an automatic system is conceived of, it should be capable of providing no errors in its working process since the absence of further controls could even arise from catastrophic consequences. Moreover, since a structure is made of infinite degrees of freedom, a large number of sensors could be necessary to monitor the target items, leading to substantial consequences in terms of costs and complexity, with a serious possibility of cancelling out the expected benefits. The fallout of these considerations is that the most of the available SHM systems are confined to well-defined structural segments (hot spots), limiting therefore their extension and their complexity, [11] or their implementation is seen as a support to human monitoring, [12,13].

The issue of an SHM system for addressing debonding, weak bonding, or no-bonding detection, should necessarily include the acquisition of structural information, which could help in distinguishing normal vs. anomalous behaviour [14,15]. Strain may be a good candidate for analysing the structural state since it is an intimate and direct measure of the material manner of working. In this sense, strain gauges are optimal options for this target, being classically employed during laboratory and flight experiments, providing the verification and validation of the structure, and permitting to carry out numerical experimental correlation and assessment of the developed theoretical models, mostly finite element-based ones [16,17]. Indeed, strain-based techniques for deformation reconstruction methods are highly diffused in literature and include, for example, the inverse finite element method (iFEM), the use of the so-called iQS4 elements for deformation data reconstruction, and finding applications for damage diagnosis and crack monitoring—as reported for instance in [18–20]—among many other works. In the first paper herein mentioned, the same authors who originally developed the iFEM method [21] also introduced the iQS4, a

four-node quadrilateral inverse element. It included hierarchical drilling rotation degrees of freedom (DOF) and was aimed at extending the capability of the method to the shape-sensing analysis of large structures. The second paper of the list exploits this technique by proposing a novel damage modelling approach where among a series of different and systematic theoretical constructions, the most suitable is selected as the one that minimizes discrepancy with respect to the response of the physical structure. Finally, in the third paper, the authors use a convolutional neural network (CNN) trained with strain mode difference data modified with random noise for estimating structural damage; the results exhibited high accuracy under different damage conditions in terms of both the localization and quantification of performance.

There are considerable drawbacks that impede the practical use of SHM: installation complexity is one of these, with the associated cabling (two or three per measuring point). Furthermore, the maintenance of such a network would be complicated and expensive. Compact sensors would therefore be preferable. The use of fibre optics reduces the amount of cables drastically and makes available a system that is immune from electromagnetic fields. They can be embedded within the structure, as largely documented in the literature, and show minimal degradation over time [22–24].

A suitable algorithm is essential for the adequate working of an SHM system to provide the elaboration of the retrieved data, which can occur offline or in real time, following the specifications of the addressed problem [25]. As a facilitator of the analysis, the correct understanding and modelling of the bonding damage effect can be useful for better addressing the research of certain characteristic features of the observed strain field [14,26]. This paper is focused on a detailed exploration of the strain field occurring on a complex composite beam after the application of a number of different loads. Such an investigation considers the strain and its derivatives on a typical surface of the reference test article, aiming at identifying the ones that better express the damage characterization. Links and comparisons with former works are discussed and exhaustive conclusions drawn.

## 2. Strain Models and SHM

In recent years, the use of fibre optics has been increasing in the field of aeronautics for the possibility of monitoring structural health via the acquisition of strain information over a large number of sensors with moderated complexity and minimal intrusiveness in terms of cabling [27,28]. Meanwhile, deformation is a precise modification of the structure under load and expresses a more direct form of correlation between external forces and the stress field. It is not a case that the most classical correlation form between numerical and experimental results passes through the comparison with strain gauge measurements [29].

Obtaining correct information to make the right decisions cannot be separated from the study of the information that is expected to be retrieved by the used sensor network. Of course, the sensor system itself contributes with its own characteristics; in this sense, the whole architecture, including the necessary electronics, should be considered since it may affect the quality of the signal. This is particularly true as far as remote connections are used, for instance for transferring the information from the aircraft to the ground station, where the main elaboration process is developed. The properties that are usually assessed when the sensor system is selected are as follows:

- Measurement range;
- Installation process and characteristics;
- Operation environment;
- Sensor sensitivity;
- Associated electronics.

Other considerations arise as a certain damage type is targeted. It is reasonable to search for its peculiar characteristics that could be disclosed by a proper acquisition network. Since the strain field describes the internal modification of a structure after a load acts upon it, this physical domain seems adequate for catching the effects of a generic flaw, which is expected to alter the original field in some way [30,31]. Once this assumption is

accepted, the second step is to investigate the variation peculiarities of these strain fields when damage, or a discontinuity, is present. In this way, this study aims to identify those structural parameters, physical characteristics, or mathematical descriptions that better identify the presence of damage. This information will be then properly used by a generic algorithm to extract the data that can indicate the presence of a flaw and consequently introduce the automatization of the process. The assessment of robust structural health monitoring algorithms, supported by smart routines of real-time data analysis and the high-fidelity data transmission, is the final aim of the illustrated process.

During the preliminary design phase, the use of numerical models and numerical analysis (for instance, FE-based), is an essential tool to verifying the correctness of the assumed hypotheses in order to evaluate their range of validity and their actual applicability to the referred problem. This kind of analysis, once the models are validated, may provide powerful insight into the microscopic behaviour the structure, offering details on the expected phenomenon and visibility of the characteristics of the effect. For instance, the region of influence of a fault can be detected, which is intended as the extension of the area where a significant strain field variation is measured [32,33]. Here, significant strain field variations are considered to be the difference that can be reasonably detected with fibre optics. For the typical systems of this kind, a suitable threshold limit ranges between 1 and 10 microstrain, which is definitely acceptable for classical structures. Results of previous works indicate that such an influence area may extend for a length that can double or triple the actual flaw length [26].

Apart from the speculative interest of such a behaviour, there is a fundamental consequence of the correct estimation of the damage extension. In fact, a flaw that extends for an area such that the structure does not enter a critical state after a limit load is usually considered to be acceptable damage, with the safety coefficient being duly considered and typically imposed at 1.5. Knowing that a certain flaw effects extend for a larger area then the measured one should lead to reduce the estimated size of the investigated discontinuity.

## 3. Test Article and Its Numerical Representation

The used test article is a representative component of a typical aircraft win -box. It is a composite beam with a rectangular cross-section. It is made of two C-type spars, symmetrical deployed, with their flanges forming the external vertical walls of the composite beam, and two flat plates, forming the top and bottom skins. The upper one has a variable thickness in order to keep the deformation value to a constant. An essential schematic is displayed in Figure 1. Production fillers are therein shown as white parallelograms. The beam elements (C-beams and plates) are all made of unidirectional CFRP material. A structural paste adhesive was applied to the caps of the C-beams so that they bonded to the cover plates.

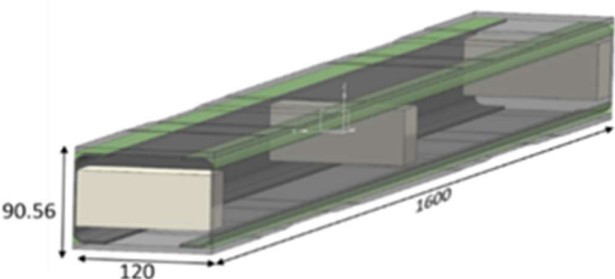

**Figure 1.** Schematic of the referred test article, a composite wing beam made of four main elements: C-beams (in green) and two cover plates (transparent). Manufacturing fillers are shown as white parallelograms [14,25].

The FE model of the referred test article is made of the following:

- Two-dimensional CQUAD elements, used for the beams wedge;

- Three-dimensional CHEXA elements for the structural adhesive, the spar caps, and the skins.

Three damage areas were introduced between one of the caps of a single spar as well as the upper skin to simulate local disbonding. The de-bonded areas, appearing as D2, D3, and D5 in Figures 2 and 3, are part of a wider experimental arrangement that includes other flaws. The consideration of only these flaws with the structure containing a larger number of anomalies is viable since damage generally produces very local effects. In other words, they vanish at a small distance from the flaws, and thus the rest of the structure is minimally influenced, if at all.

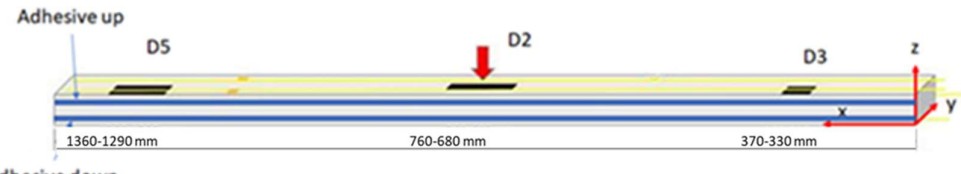

**Figure 2.** Schematic of the referred test article, with the imposed damage locations, position of the symmetric concentrated external load, and the adopted reference system.

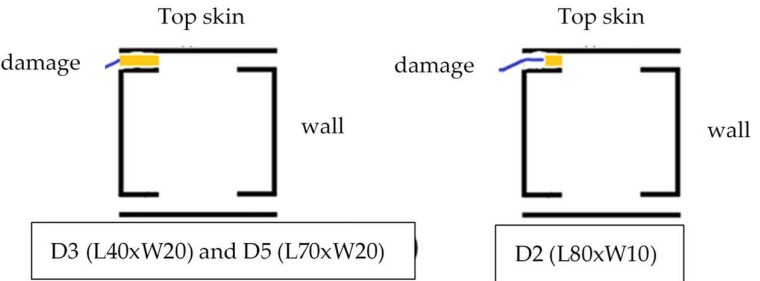

**Figure 3.** Test article section with details on the flaws' extension both in the longitudinal direction (text reported under the picture) and along its width (orange rectangle in the sketch). Numbers report the measures in mm. L—length; W—width.

These imposed flaws can be experimentally realised by interposing a Teflon layer between the spar caps and the upper plate in the designated regions. In the simulation, such discontinuities were replicated by imposing a very small elastic modulus to the relevant material. Previous experiences, carried out by other authors [14,26], indicated that this approach was equivalent to the removal of the corresponding FE. In the Figure 4, a parametric study of the effect of Young's modulus reduction is shown (the results corresponding to element removals have been fictitiously reported as a Young's modulus reduction of $1 \times 10^{16}$, and the same results obtained for a reduction of $1 \times 10^{6}$ up to four significant digits are shown).

The modulus reduction approach implemented in this study is largely faster in terms of modelling and safer from the point of view of the risk of running into errors. The general length density of the mesh is equal to 8 mm, but it was made denser in the flaws area, with the aim of obtaining the best resolution in terms of strain description and attention to the damage effect in terms of strain, which was expected to be larger than the actual discontinuity; see Figure 5. In order to prevent change of the mesh density from affecting the strain outcome in the de-bonded regions, the flaw area was extended to 5 times the damage length for a total extension of 350 (D5), 400 (D2), and 200 (D3) mm, respectively. Additionally, simulations with varying mesh densities (namely 0.5, 1.0, and 8.0 mm) were performed: after realizing that the results with mesh densities of 1.0 mm and 0.5 mm were close to each other, we chose the model with a mesh density of 0.5 mm for the subsequent strain investigations.

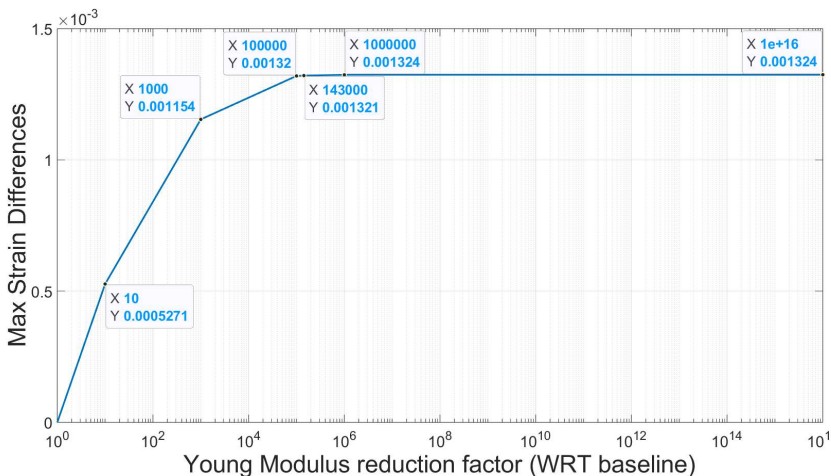

**Figure 4.** Parametric study of the effect of Young's modulus reduction to simulate debonding.

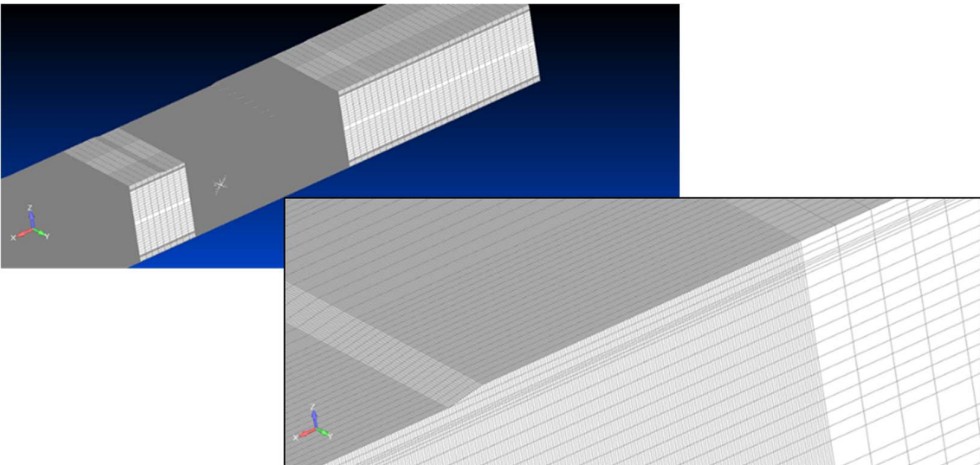

**Figure 5.** A representation of the FE model of the referred test article (**top**) with details on the mesh refinement (**bottom**).

The considered constraints simulate typical three-point bending tests. Supporting regions are schematized close to the beam extremities, allowing rotations, but inhibiting positive and negative vertical translations. In this configuration, as a representative load arrangement, a symmetrical excitation was chosen, consisting of a perpendicular concentrated force acting at the centre of the beam and spread along the width. The resulting deformed shape is shown in Figure 6.

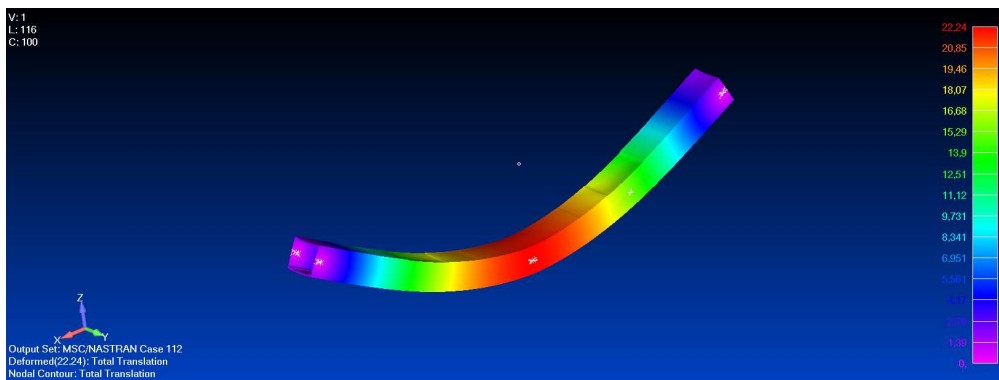

**Figure 6.** Deformed shape of the reference beam, supported simply, under a symmetrical load.

### 4. FE Model and Strain Computation

The abovementioned FE model was run in the undamaged and damaged configurations for the referred load condition in order to retrieve strain data to characterize the damage effects on the structural response. In particular, aiming at the implementation of an optical fibre system based on the use of FBG, deformations were considered along a specific direction, spanwise, simulating the presence of a 1-directional sensor array. The strain field was therefore numerically calculated along the X-axis, $\varepsilon_x$, congruent to the beam length extension, and the Y-axis, $\varepsilon_y$, associated with the transversal direction over the top plate surface. These two quantities are calculated by a direct elaboration of the beam displacements along the corresponding directions ($x$ and $y$), according to the following strain definition:

$$\varepsilon_{x,y} \overset{\text{def}}{=} \frac{\Delta l_{x,y}}{l_{0x,y}} = \frac{l_{x,y} - l_{0x,y}}{l_{0x,y}}$$

where $l_{0x,y}$, $l_{x,y}$, and $\Delta l_{x,y}$ are the original length along the X and Y axes, the corresponding length modified after the action of the load system, and their variation, respectively. An illustration of the implemented concept is shown in Figure 7.

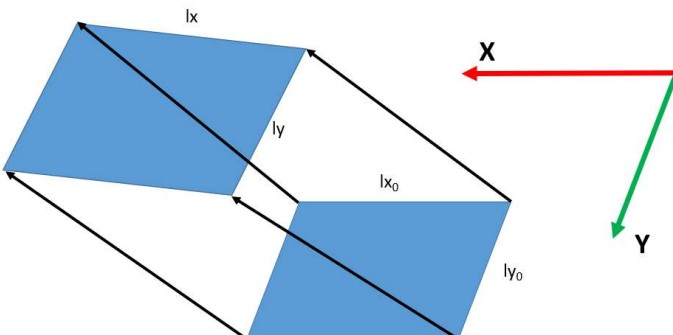

**Figure 7.** Parameters involved in the computation of the strain along the X and Y directions.

Since the FBG strain sensors may be used to measure the deformations in the damage and undamaged conditions, such data may be used to evaluate their variations and to provide information regarding the damage effects on the structural response. Therefore, the quantities $\Delta\varepsilon_x$ and $\Delta\varepsilon_y$ were evaluated, together with their direct derivatives, i.e., $\frac{\partial\Delta\varepsilon_x}{\partial x}$, and $\frac{\partial\Delta\varepsilon_y}{\partial y}$, to highlight the regions where the largest strain variations occurred. The results for the symmetrical load are shown in the following paragraph, as defined above. All the results refer to the strain field along the X and Y directions, revealed on the external surface of the top plate.

### 5. Strain Analysis

The elaborations are reported in the figures below, ranging from Figures 8–13. First, the strains evaluated in the longitudinal (spanwise) and transversal directions (along the width) are reported in Figures 8 and 9. It is immediately discernible that the strain field is mainly affected by the sharp variability of the thickness as clearly indicated by the singularity lines appearing on the diagrams corresponding to the thickness steps. Such discontinuities can be avoided by acquiring the deformations in the inner part of the structural system and specifically in the bonding layer or even in the internal components (for instance, at the bottom side of the cap). In this case, however, it was chosen a challenging region to allow some peculiarities of the strain field to arise. The scale is not reported in the graph, but it is sufficient to state that the values are in the order of thousands of microstrain under a load that is representative of the actual operation of the composite beam herein investigated.

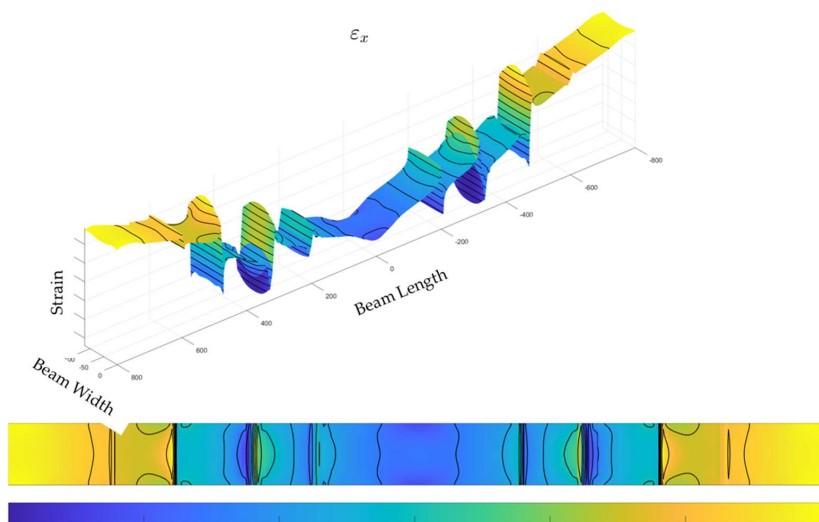

**Figure 8.** Strain along the X-axis.

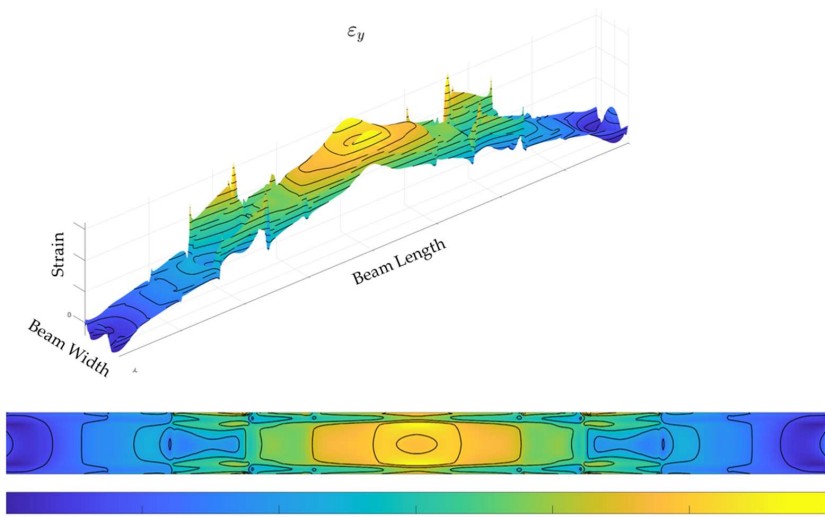

**Figure 9.** Strain along the Y-axis.

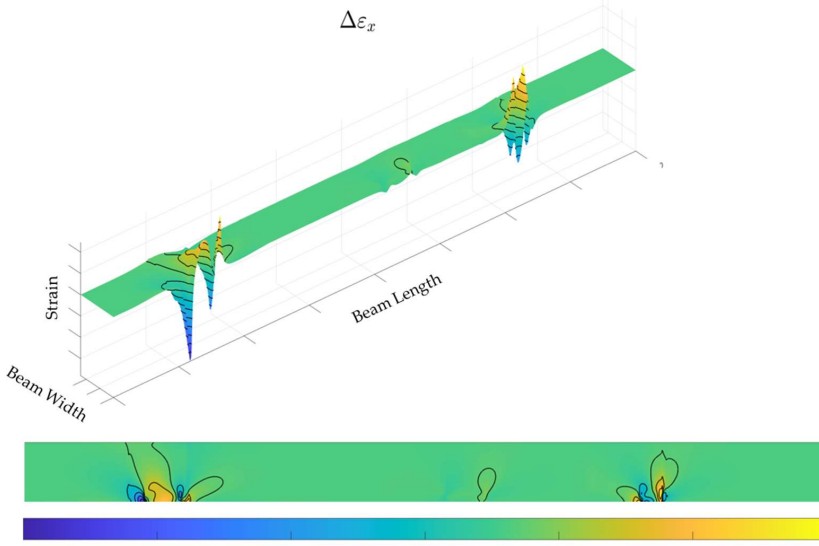

**Figure 10.** Strain along the X-axis, difference between damage and undamaged configuration.

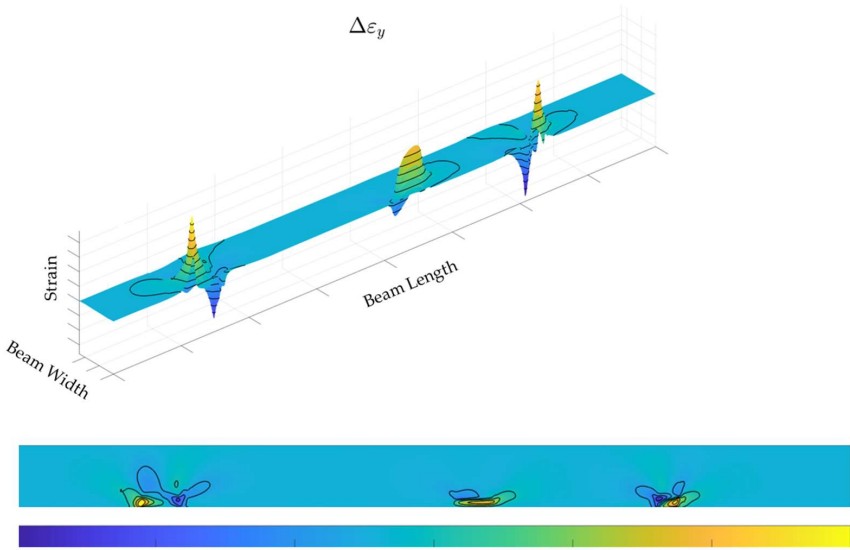

**Figure 11.** Strain along the Y-axis and the difference between the damage and undamaged configuration.

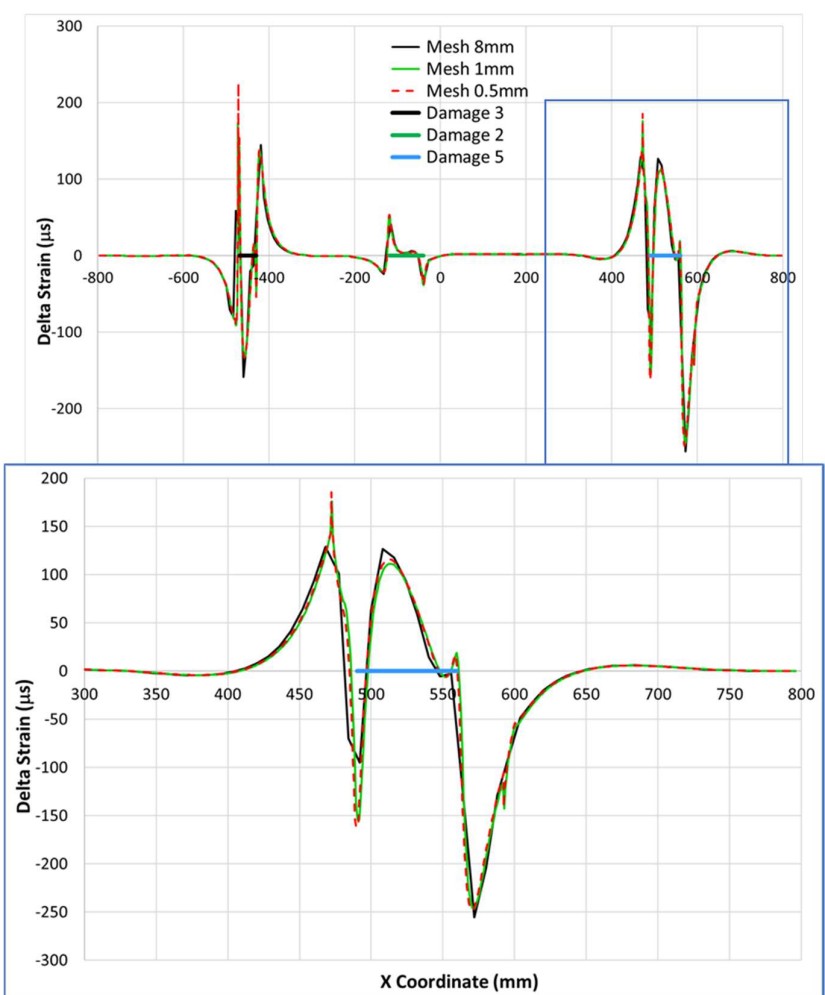

**Figure 12.** Comparison of delta strain for three dimensions of the model mesh (8, 1, and 0,5 mm) with a focus on damage 5 (blue rectangle).

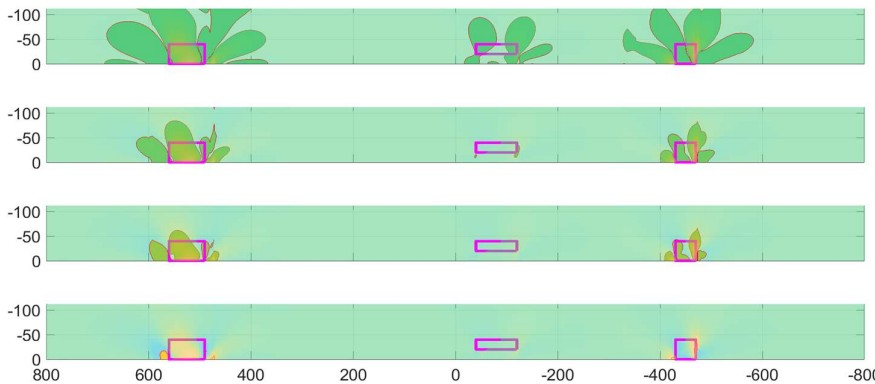

**Figure 13.** Strain variation ($\varepsilon_x$) between the damaged and undamaged conditions, with the evaluation in the top panel, which was then filtered by cancelling the values below an established threshold and fixed at 10, 50, 100, and 350 µs (top to bottom). In the pictures, the actual flaw boundaries are reported in violet.

The graphs are very similar in the damaged and undamaged case, so it is difficult to appreciate significant variations between the two cases. However, there is a difference in the structural response in terms of strain between the damaged and undamaged case (indicated as $\Delta\varepsilon_x$ and $\Delta\varepsilon_y$, respectively, for the difference between the strain along *x* and along *y*, in the damaged and undamaged configurations), as seen in Figures 10 and 11, giving rise to the following observations:

- The structural discontinuity effects (i.e., the contributions linked to the thickness steps), disappear. This can be considered a trivial result since they are present in both the circumstances analysed in this study, which does not change in the two configurations (damaged and undamaged). However, it does also mean this kind of imposed damage (debonding) does not affect the structural response macroscopically.
- More generally, the entire structural response flattens, and only the damaged regions emerge. This outcome is consistent with the first one, and it should indicate that the flaws generate a very local effect, which does not extend outward much from the occurrence area. While this may appear to be simple confirmation of other results already reported in literature, this new representation highlights how the discontinuity propagates along the top panel, albeit in a very limited way. In this case, such variations may be measured, in the analysed case, around hundreds of microstrain (i.e., a magnitude less than before). The values are nonetheless far from the bottom threshold of usual fibre optics which should not go under tens of microstrain (i.e., a further magnitude below).

It may be concluded that both $\Delta\varepsilon_x$ and $\Delta\varepsilon_y$ are significant representations of the damage effects in the structure, with significant peculiarities:

- $\Delta\varepsilon_x$ provides the highest values, useful for detecting the damage areas and possibly monitoring their growth; however, these values seem to be concentrated at the extremities (structural discontinuities), therefore highlighting a very limited possibility of catching the right points for a robust identification;
- $\Delta\varepsilon_y$ provides a more homogeneous deformation pattern, somehow complementing the effectiveness of the former detection, increasing the possibility of damage detection, and so compensating for its lower numerical significance.

As an expansion of the previous comments, the actual possibility of deploying the optical sensor array in the desired directions and along the desired path in a real application should be considered. Specifically, in the case of a beam like the one considered here, it could be very difficult to arrange a fibre in the width direction, and so a specific manufacturing processes should be designed and implemented.

These preliminary results necessitated further investigations for establishing the correctness of the adopted model and most importantly, for assessing the validity of the mesh. In other words, a preliminary convergence study process was assessed and is presented here.

## 6. Model Assessment

One of the most direct ways to verify the goodness of a numerical model is to check how a variation of the adopted mesh causes the modification of the results. Then, the mesh density is increased by about a magnitude by implementing refinements characterized by a 0.5- and 1.0-mm step. For the sake of the model complexity, the augmented density of the composite beam numerical representation is limited to the area around the imposed flaws for an extension equal to five times the original length. This choice was informed by a previous result [26], which indicated how the damage effect extends to a region even 2–3 times larger than its size for the considered dimensions. Therefore, a conservative approach was followed.

The results in terms of strain variations obtained by the FE models with different mesh densities (8, 1, and 0.5 mm), are shown in Figure 12. The very first impression is that the outcomes are very similar. This is very important in the sense of confirming the reliability of the outcomes. However, deeper analysis reveals that the finest meshes yield the smoother results; furthermore, differences between the 1- and 0.5-mm step models are barely appreciable. Then, it is reasonable to state that the FE numerical model is approaching its convergence at these levels of detail. For the sake of clarity, it should be stated that these specific analyses were carried out for uniform meshes, extended all over the beam, while further investigations considered a refinement limited to the damaged regions.

Other considerations arise with a closer observation of the strain functions in the different cases. The 8 mm mesh leads to emergence of four small, but still evident, peaks. All such marked discontinuities correspond to structural irregularities, whether in terms of skin thickness variations or at the beginning and the end of the flaw. In turn, this indicates that a non-appropriate mesh could exacerbate the effects of any kind of variation in the considered architecture, which could lead to modification of the local stiffness (directly affecting the local strain values). This outcome is in line with the well-known characteristic of the discretization models leading to local, unrealistic peaks in the proximity of holes, for instance. In this case, it should be also underlined that the mesh was ten times denser than the typical flaw dimension in the considered direction, and that this is not sufficient to obtain reliable results.

The differences were computed to be around 20–30% at the discontinuity regions, and were almost non-existent elsewhere. Such variations seem more evident for the flaws that are deployed near to the thickness variation zones (namely, D3 and D5), than they are for damage in a plain zone (D2). Finally, the complex strain function that was obtained for D3 and D5, with respect to what was achieved for D2, may be at least partially attributable to a sort of interaction between the two concurring discontinuities. It should be noted that since a linear analysis was conducted, the strain levels did not affect the strain function shape; furthermore, another concurrency factor may be found in the strain extraction on the external skin surface, where the deformation differences are expected to be larger (as the skin itself is irregular).

## 7. Discussion

The ability to detect a damage is a function of the induced deformation pattern and its corresponding levels. These parameters should be compared to the minimum threshold that the monitoring system is able to detect. Then, it is reasonable to investigate how the strain difference outlines, such as for the instance proposed in Figures 10 and 11, and modifies the graphs by introducing a minimum recognizable level. In Figures 13–16, such a proposed process is applied to the strain variations between the damage and undamaged

conditions $\varepsilon_x$ and $\varepsilon_y$. There, the areas of the beam surfaces are highlighted, where the strain variations exceed some established thresholds fixed at 10, 50, 100, and 350 microstrain.

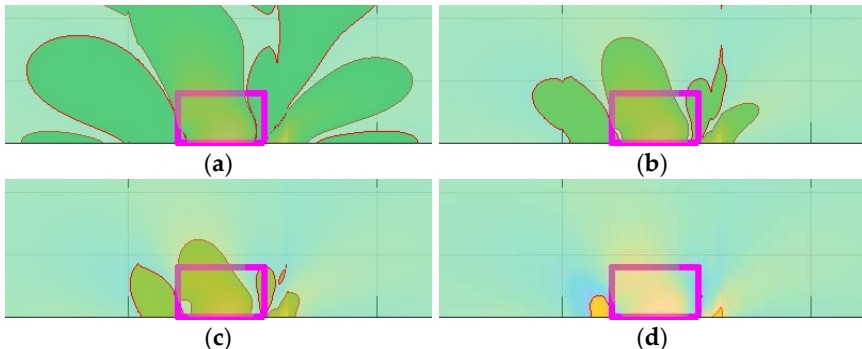

**Figure 14.** Focus on $\Delta\varepsilon_x$ for damage 5 at levels: (**a**) 10 µs, (**b**) 50 µs, (**c**) 100 µs, and (**d**) 350 µs.

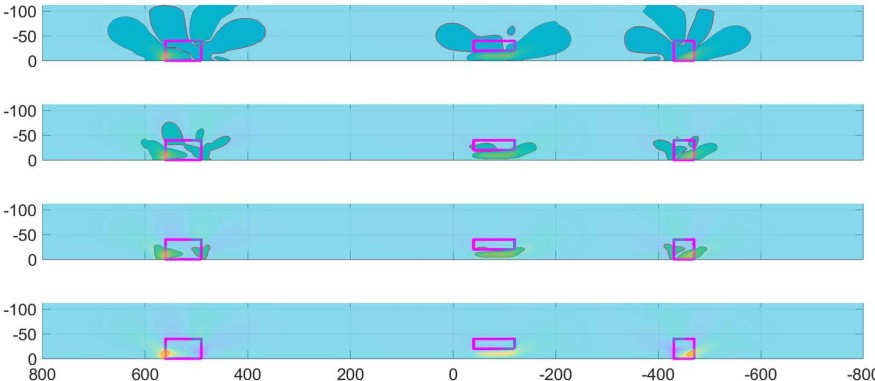

**Figure 15.** Strain variation ($\varepsilon_y$) between the damaged and undamaged conditions, with the evaluation on the top panel, which were then filtered by cancelling the values below an established threshold and fixed at 10, 50, 100, and 350 µs (top to bottom). In the pictures, the actual flaw boundaries are reported in violet.

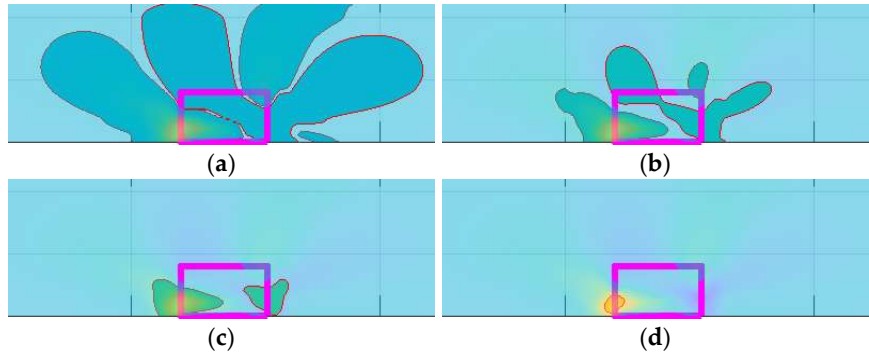

**Figure 16.** Focus on $\Delta\varepsilon_y$ for damage 5 at levels: (**a**) 10 µs, (**b**) 50 µs, (**c**) 100 µs, and (**d**) 350 µs.

As expected, the detectable area drastically reduces as the threshold increases. If this phenomenon decreases the probability that a monitoring system is able to point out the existence of a flaw, conversely, it also restricts the observable region to the actual damage zone (i.e., the area interested by a structural discontinuity), therefore augmenting the resolution of the tool. It is important to observe that such a process should be handled with extreme care since some weaker damage (such as with D2) is shown almost to vanish, even for small reductions of the threshold. This one, however, may only be a minor drawback since less relevant flaws have limited importance in the evaluation of the structural health, or equivalently, in the estimate of residual strength. Surprisingly enough, such an aspect

is less evident for the spanwise deformation ($\varepsilon_x$) than for the strain measured along the transversal direction (along the width, $\varepsilon_y$).

To have a deeper view of the phenomenon here described, a focus on a selected and representative damage area is reported, D5, for both of the in-plane deformations. In Figure 14, $\varepsilon_x$ is first considered. The first aspect that arises is a sort of a multilobe effect due to the presence of the damage to the surrounding area, which extends all over the plate up to the other extremity. Therefore, the localization of the flaw effect should be considered within this limit. It is also apparent that as such an effect is very small, it is drastically resized as the strain detection window is lowered to just under 50 microstrain, which can be considered a first conservative, highly observable value. The highlighted area shrinks almost conformally, i.e., keeping its original shape. This process continues in the same way as the limit level is brought to 100 microstrain: the shape appearance of the flaw effect on the strain field is kept almost constant but is slightly reduced. Finally, when the observability threshold is moved under 350 microstrain (an arbitrarily selected value), it can be seen that the relevant effects concentrate only on the borders of the interested area, which represent the discontinuity lines. This behaviour is repeated almost equally for the other strain component, $\varepsilon_y$; see Figure 16. Some difference may be appreciated in the lower absolute values, as the detected areas are smaller for the considered observability levels. Although the effect zones keep their shape as the process moves on, it seems that this redesign converges towards the flaw borders faster than before, as mentioned above. In particular, already at the 100 microstrain window, the effect is concentrated on the flaw area limits, while at 350 microstrain, a single lobe appears on the first edge.

A synthesis of such an analysis is reported in Figure 17, where a diagram is shown, portraying a parameter representative of the detectable area associated with the Y-axis, and the imposed visibility threshold, associated with the X-axis. This figure is particularly important because it summarises the occurrences for all the three considered flaws, while the former analysis is limited to the evidence for D5 only. The parameter indicative of the detectable area is synthesized as the ratio between the extension of the region along which a significant strain variation occurs as a consequence of the flaw presence with respect to the undamaged configuration and the actual extension of the flaw. This suggested process reduces the areas to a nondimensional parameter, which allows for the representation of all the imposed irregularities on a single picture. There, the horizontal blue line identifies the original dimension of the fault; i.e., when the ratio approaches 1, this means that the detectable area has the same size of the original flaw (its shape could be different however). Such a unit value could be considered, in a first approach, as the limiting acceptable level of the detection system or the one capable of identifying at least the actual flaw measure. The reported graph in Figure 17 displaces a logarithmic scale for the X-axis, while a classical linear scale is associated with the Y-axis. For the sake of clarity, investigated thresholds have been drawn as vertical thin blue lines.

It may be observed that, for ideal system capabilities (i.e., threshold levels under 10 microstrain, a good performance not very far from that obtained with available commercial systems), all the imposed flaws may be detected with a large margin. The selected characteristics however collapse as such a threshold approaches 100 microstrain. As expected, the curve describing the damage D2 is the one that moves more rapidly towards the assumed detection limit (unit horizontal line, blue); however, confirming the previous observations, such a behaviour is more evident for $\varepsilon_x$ than for $\varepsilon_y$. Irrespective of the relative sizes, the tendency is more homogeneous for the deformations along y (flaws D2, D3, and D5), while it is noticeably different for the strain along x (the D2 characteristic is well beneath the others). In other words, the two all-width flaws (D3 and D5) have a similar behaviour, while D2 is distinct. At the fixed level of 100 microstrain, the following occurrences are found:

- D2, $\varepsilon_x$, well under the assumed identifiability limit;
- D2, D3, and D5, $\varepsilon_y$, at the limits of the assumed observability threshold;
- D3, and D5, $\varepsilon_x$, well over the assumed detectability level.

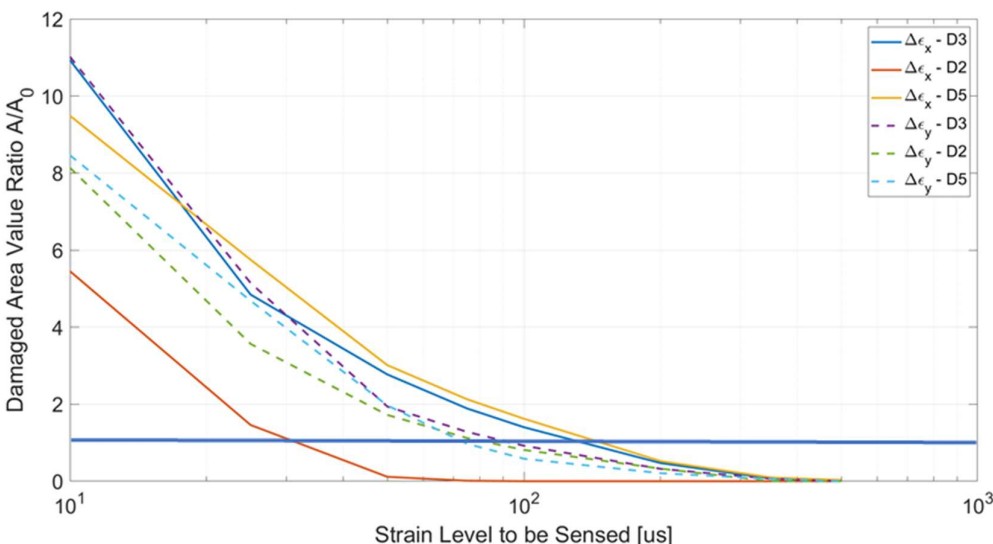

**Figure 17.** Damage ratio vs. (assumed) detectable threshold level.

It should be once more stated that these results take into consideration the numerical values, exclusively, and not the technological applicability: it may be somewhat difficult to deploy a sensor array along a typical beam width for geometrical constraints. In these cases, a dedicated installation process should be investigated and possibly assessed.

Finally, it seemed interesting to compare specific results of the present study with the outcomes of a previous analysis on the same beam, [26], examine the relation between damage effects length and actual flaw length for all-cap width flaws. In that work, a function was extracted, relating the abovementioned parameters on an 8 mm mesh. It was therefore necessary to synthesize a similar parameter herein, which could reduce the dimension-order of the 2D scalar characteristics used in this study (the flaw region area). It was chosen to multiply the square root of the ratio between the calculated and nominal areas (a non-dimensional 2D peculiarity) by the nominal length of the damage. In this way, a featured damage length could be obtained, comparable to the one-dimensional value of the referenced analysis.

In Figure 18, the dots represent the three analysed flaws for flaw detection thresholds equal to 25 μs, while the continuous red line is the function evaluated in a former analysis [26], linking the flaw length and the flaw effect length. In this figure, the nominal damage lengths are reported along the X-axis, while the synthetic damage effect length is reported along the Y-axis. It is worth noting that the function represented in Figure 18 (red line) was constructed via reference to a parametric study focused on a specific position of a certain damage on the same reference structure; in the current article, three different flaws were analysed instead, each located in the specific places of the test article, and one of them exhibiting a remarkable geometric variation (it extended for just a half-width of the cap). Therefore, it is not surprising that the data are not perfectly matching; instead, the not-so-distant variations with respect to the original line are sufficiently encouraging for further studies. This correlation was meant to be a first step towards producing a possible enhancement of this estimated function.

A good agreement is obtained, which seems to break only for the D2 flaw, as expected (since the previous study considered only full-width discontinuities). In spite of the very limited comparison, this evidence permits the conclusion that the process seems to produce coherent values with an independent investigation. Additional tests are necessary to further prove this preliminary result, which could also be directed to test the validity of the introduced 1D synthetic descriptor which could have a significant relevance in further studies.

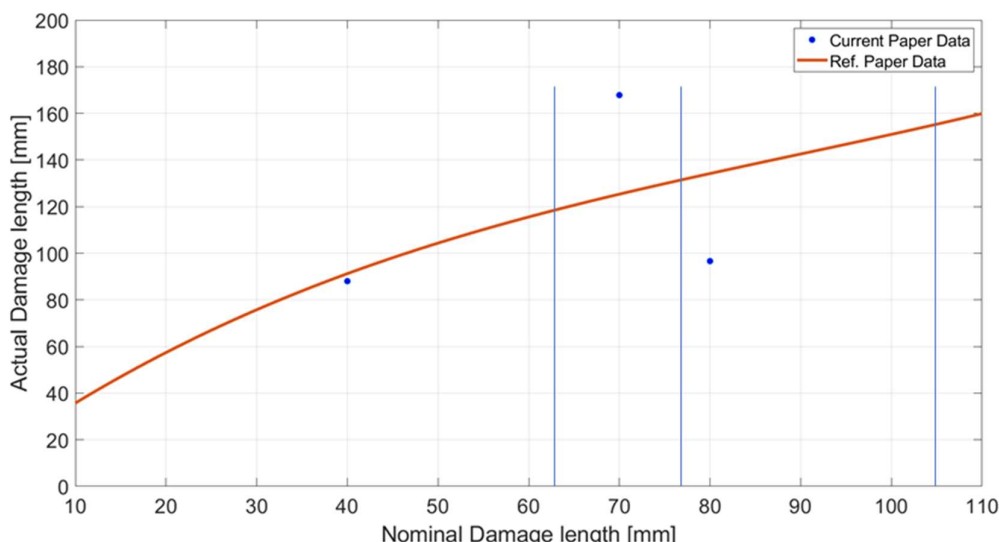

**Figure 18.** Damage effect length vs. Nominal Damage length.

### 8. Conclusions and Comments

The paper reports a study aimed at the numerical characterization of the adhesive damage effects on a typical composite beam, made of four main components bonded together. The test article was made of two symmetric C-shaped spars, forming the external wall of the considered specimen, covered by two plates making the top and bottom skins. Adhesion occurred between the spar caps and the interested parts of the covering panels. The result was a rectangular section beam. Top panel was tapered, i.e., characterized by a variable thickness along its span. The damage effects were determined in terms of strains, with the deformation field one of the most salient expressions of the structural response under a generic load or external force. The focus was on the strain field differences between the damaged and undamaged conditions: in fact, the overall structural response was not observed to change markedly in the two configurations; however, when the structural responses were compared, relevant strain differences did appear but only in the region of the flaws. They were a magnitude less than the absolute values but significant enough to emerge clearly and well above the observability of common sensing systems: for $10^3$ microstrain absolute structural responses, the variations could be evaluated in about $10^2$ microstrain. The following main conclusions may be drawn:

- The study confirms that the effect of a discontinuity in the structural architecture of a generic component is mainly local, i.e., it does not extend very far from its occurrence. As such, it is particularly hard to detect, which implies that the monitoring network should be dense.
    - In particular, the investigation focused on local debonding, simulated through a deterioration of the elastic characteristics of the representative elements of the adopted FE model in some established zones.
    - Imposed irregularities effects were observed in the upper surface of the beam, which was separated by the debonding areas through the panel itself. It was discovered that the presence of the thickness discontinuities affects the strain response, somehow superimposing their influence on the structural response to the ones caused by the presence of the flaw.
    - The result is a very complex deformation function, extending well past the regions of the flaw.
- The analysis allows suggests that a medium mesh (8-mm step) may well catch the phenomenon, as a much finer discretization does not significantly change the strain variation predictions.

- o In fact, the 0.5 and 1mm step meshes largely confirm the preliminary results, quantitatively.
- o Significant modulations may, however, be observed at the discontinuity zones (20%), both at the stations where thickness variations occur, and at the boundaries of the imposed flaws. Such variations do extend for a few mm, much less than the damage size, which was fixed to 102 mm in the investigation.
- o From the 2D strain maps, the top plate shows interesting diffusion of the damage effects over all the width, extending the flaw size further and in a spanwise direction as well.
- o The application of a filter indicates that that such an extension is true for very small values of the strain difference, so that the area is more than halved by moving from a 10 to a 50 microstrain threshold. Higher values then appear and concentrate only at the flaws' boundaries.

- Based on the above considerations, a parametric study was conducted, which ultimately led to an understanding as to how much the observability of the damage may be conditioned by the capability of an idealized sensing system.
  - o For this, a nondimensional parameter was introduced, defined as the ration between the area where the damage produces its effect and the nominal damage area.
  - o It was observed how, for a 100 microstrain limit deformation observability level, the affected area reduces to the nominal flaw area, which in turn may be assumed to be the minimal acceptable resolution for a generic health monitoring system.
  - o Under this level, the damage almost disappears from the monitors. Irrespective to the maximum values, extremely limited effects should be barely visible from a generic instrument, and this occurrence may be easily classified as a disturbance or system error.

- The study reveals that the deformations along the spanwise and the width direction provide very similar results, with the latter potentially offering some interesting insights into the flaw effects.
  - o Even with this possibility, there may be some difficulty in realizing this for practical applications, and it should be considered for further developments.
  - o Its characteristics are extremely interesting for width-limited flaws since its persistence at different observability levels is much more significant than is the one in the nominal, spanwise direction. This topic should be examined in the subsequent steps of research.

- Despite its limitations, the paper provides interesting information for characterizing damage effects and outlines important criteria for designing appropriate sensor networks intended to detect the presence of flaws.
  - o With reference to a specific region of a basic test article (its upper surface) that nonetheless contains all the important characteristics of a general composite structure, the study reports how the strain field modifies as a consequence of the deployment of debonding regions with respect to the original map.
  - o Therefore, our study indicates how it could be possible, in principle, to have direct evidence for the presence of insurgent defects by comparing the nominal vs. the current deformation distribution.
  - o In turn, this would enable a continuous evaluation of the strain field modifications vs. a reference state, which would be representative of the system integrity. Such a step is not trivial and would in turn offer the possibility of assuming a manufacturing process with high quality levels. In other words, it should ensure the similarity of all the released products.

Concerning the future developments of the investigations, the following considerations hold:

- The impact of structural discontinuity on the strain field variation and the importance of the mesh at the flaw boundaries should definitely be explored further in future studies.
- Similar, the impact of the mesh on the structural response variations at the edges should also be better understood, in order to better calibrate the FE models.
- Another essential element is the characterization of the debonding effect along the thickness of the beam. Previous studies did in fact show how such an effect may be modulated transversally and how the strain variation is modified along the section; a systematic assessment is still missing, however.
- Although all the data were properly and extensively collected, further experimental series should be completed before any final conclusion is drawn, specifically aimed at validating the proposed models and the preliminarily outcomes reported here.

**Author Contributions:** Conceptualization, L.P., A.C. and S.S.; methodology, G.D., A.S., L.P., A.C., G.A. and D.B.; software, G.D. and A.S.; validation, G.D. and A.S.; formal analysis, G.D., A.S., L.P. and M.C.; investigation, L.P. and A.C.; resources, A.C. and I.K.; data curation, G.D., A.S. and L.P.; writing—original draft, G.D., A.S., L.P. and A.C.; writing—review and editing, G.D., A.S., L.P., A.C., M.C., G.A., S.S., I.K. and D.B.; visualization, G.D., A.S. and L.P.; supervision, L.P.; project administration, A.C. and I.K.; funding acquisition, A.C. and I.K. All authors have read and agreed to the published version of the manuscript.

**Funding:** This research was funded by RESUME under contract N.950, 4 December 2020, Italian Ministry of Defense—Secretariat General of Defense and National Armaments Directorate (SGD/DNA).

**Data Availability Statement:** Data are not available publicly for reasons of confidentiality.

**Acknowledgments:** The results presented here were obtained as part of the abovementioned project, RESUME (Real-Time Structural Health & Usage Monitoring Systems for UAV). The authors gratefully recognize the continuous and appreciated support of the staff of SGD/DNA.

**Conflicts of Interest:** The authors declare no conflict of interest.

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
