# Peer review of "Numerical Analysis Results of Debonding Damage Effects for an SHM System Application on a Typical Composite Beam"

_aerospace, doi:10.3390/aerospace10060507_

Round 1

Reviewer 1 Report

The paper is written well overall and be publishable if necessary revisions are made. Here is the list of them:

First, the literature survey can be extended by including other strain-based deformation reconstruction methods, e.g., inverse finite element method (iFEM), application of iQS4 elements for deformation/strain data reconstructions, damage diagnosis and crack monitoring inverse finite element methods, etc.

Secondly, the methodology is based on the FEM simulated strains data. To improve the mathematical aspect of the paper, damage analysis strategy/algorithms can be provided with the finite element equations of the plate/shell elements.

Thirdly, in Section 3, damage modeling/debonding can be discussed further. The authors mentioned that debonding simply can be modeled as providing a low elastic modulus to the related region. Is this always true for different directional loads? With this assumption, can we make a direct static analysis? what should be degradation factors?

Lastly, the authors can explain the large differences between the reference and current damage length predictions observed in Figure 17 in a better way.

Finally, I believe that all Figures' quality can be improved. Instead of plotting figures in Excel, the authors can use better plotting options, such as OriginLab, Matlab etc. The size of the text in the figures can be arranged in a similar size to the font size of the main text of the paper.

Reviewer 2 Report

It is an original paper dealing with “Numerical Analysis Results of de-bonding damage effects for a SHM System Application on a Typical Composite Beam “.Regarding this manuscript there are some minor and major comments below to help the readers to be more beneficial from the paper.

1.      The abstract is written as a general description. It is worthwhile to describe your main achievements, and results

2.      In line 54, the authors describe the de-bonding as a type of damage in composite materials in industry without any references

[a] Effects of nanoparticles on nanocomposites mode I and II fracture: A critical review. Progress in Adhesion and Adhesives, 3, 391-411

3.      In line 62, what is NDI? It is recommended to write non-destructive inspection

4.      In line 62, The authors need to have an indication to the most important of NDI methods in composite structures by addressing the following references

[b] An improved method of eddy current pulsed thermography to detect subsurface defects in glass fiber reinforced polymer composites. Composite Structures, 242, 112145.

[c] Non-destructive testing and evaluation (NDT/NDE) of civil structures rehabilitated using fiber reinforced polymer (FRP) composites. In Service life estimation and extension of civil engineering structures (pp. 193-222). Woodhead Publishing.

[d] Damage detection of CFRP composites by electromagnetic wave nondestructive testing (EMW-NDT). Composites Science and Technology, 210, 108839

5.       What is the advantage of SHM over the NDI technique? There are a lot of SHM methods, the authors need to bring the most important of SHM methods to clarify the different between NDI and SHM methods by addressing the following references

[e] Impedance analysis for condition monitoring of single lap CNT-epoxy adhesive joint. International Journal of Adhesion and Adhesives, 88, 59-65.

[f] Characterisation of local damage in pultruded GFRP road bridge decks with random fibre mat misalignments. Composites Part A: Applied Science and Manufacturing, 152, 106673

[g] Structural health monitoring of adhesive joints under pure mode I loading using the electrical impedance measurement. Engineering Fracture Mechanics, 245, 107585

6.      How the mesh sensitivity was evaluated in this study

7.      In general, the damage in composite structure would be made due to damage extension and stress concentration. Why have the authors not used numerical techniques to illustrate the stress distribution?

8.       Use bullets in Conclusions to emphasise the main achievements of the paper

Minor editing of English language required

Round 2

Reviewer 1 Report

The revision of the paper is done well by the authors. However, there is one single revision and correction on the references required before publication. The revised sentence, "Indeed, strain-based techniques for deformation reconstruction methods are very diffused in literature, for instance involving inverse finite element method (iFEM) involving the use of iQS4 elements..." missed original study of the iFEM/iQS4 (a quadrilateral inverse-shell element with drilling degrees) element published in 2016. Moreover, the authors did not include the damage detection studies regarding iFEM for assessment of impact damage and non-trivial boundary conditions published in 2021. Also, automatic crack size estimation with iFEM published in 2023 and the original study of coupled iFEM and peridynamics formulation for crack propagation monitoring published in 2022 are missed in the relevant sentences. Therefore, the authors are requested to provide the correct references and make proper revisions on the related sentences in the introduction.

Reviewer 2 Report

1. In response to comment 2 (In line 54), why the reference [a] "Effects of nanoparticles on nanocomposites mode I and II fracture: A critical review. Progress in Adhesion and Adhesives, 3, 391-411." is not included in the modified version.

2. Why the authors name in ref [8] and [9] are not included in the modified version.

Round 3

Reviewer 2 Report

Accept in present form